# The Impact of Long-Term Care Home Ownership and Administration Type on All-Cause Mortality from March to April 2020 in Madrid, Spain

**Maria Victoria Zunzunegui** [1,*][ID]**, Manuel Rico** [2]**, François Béland** [1] **and Fernando J. García-López** [3]

1 École de Santé Publique, Université de Montréal, Montreal, QC H3N 1X9, Canada; francois.beland@umontreal.ca
2 infoLibre, 28010 Madrid, Spain; manuel.rico@infolibre.es
3 National Epidemiology Centre, Carlos III Health Institute, 28029 Madrid, Spain; fjgarcial@isciii.es
* Correspondence: maria.victoria.zunzunegui@umontreal.ca; Tel.: +34-692-064-134

**Abstract:** Our aim is to assess whether long-term care home (LTCH) ownership and administration type were associated with all-cause mortality in 470 LTCHs in the Community of Madrid (Spain) during March and April 2020, the first two months of the COVID-19 pandemic. There are eight categories of LTCH type, including various combinations of ownership type (for-profit, nonprofit, and public) and administration type (completely private, private with places rented by the public sector, administrative management by procurement, and completely public). Multilevel regression was used to examine the association between mortality and LTCH type, adjusting for LTCH size, the spread of the COVID-19 infection, and the referral hospital. There were 9468 deaths, a mortality rate of 18.3%. Public and private LTCHs had lower mortality than LTCHs under public–private partnership (PPP) agreements. In the fully adjusted model, mortality was 7.4% (95% CI, 3.1–11.7%) in totally public LTCHs compared with 21.9% (95% CI, 17.4–26.4%) in LTCHs which were publicly owned with administrative management by procurement. These results are a testimony to the fatal consequences that pre-pandemic public–private partnerships in long-term residential care led to during the first months of the COVID-19 pandemic in the Community of Madrid, Spain.

**Keywords:** mortality; long-term care; COVID-19; public–private partnership; Spain

## 1. Introduction

During the first months of the COVID-19 pandemic, international variations in mortality in long-term care homes (LTCHs) were related to differences in the pre-pandemic elderly care sector structure and organisation and in the overall public health response to the pandemic [1]. In Spain, the high mortality in LTCHs highlighted the widespread inability to protect lives [2].

Understanding whether LTCH characteristics can explain the mortality rates during this public health emergency is crucial to improving pandemic preparedness and the quality of long-term care. A review of the association between LTCH characteristics and COVID-19 mortality found that the strongest and most consistent predictors of COVID-19 deaths were the size of the LTCH; whether it was located in an area with a high prevalence of COVID-19; lower staff-to-resident ratios; and a lack of staff unionisation [3]. In Canada, government-run LTCHs had lower mortality than for-profit LTCHs [4,5]. For-profit LTCHs had the following organisational factors that are associated with higher numbers of COVID-19 deaths among residents: they were more often crowded, and their staff were more often underpaid, less likely to secure full-time positions, often employed at more than one center, or employed on casual terms with fewer benefits when compared with staff at nonprofit private or public homes [6–9]. Further studies conducted in Australia, England, and Spain confirmed that larger homes experienced higher mortality rates; however, the findings on

the impact of ownership and administration type were inconsistent. In Australia, nonprofit providers operating multiple homes had higher COVID-19 case fatality rates [10]. In England, where the care home market is mostly private, excess mortality at LTCHs was associated with larger operators, while no association was found with the profit status of the provider [11]. In Spain, higher COVID-19 mortality was found in LTCHs under private ownership, but this study did not examine any PPPs [12]. While the specific findings of all the above studies are context-dependent, they point to several LTCH characteristics as leading factors in increasing or decreasing the likelihood of mortality during a public health emergency.

In Spain, public universal coverage of long-term care was established in 2006 when legislation to provide long-term care was passed in order to provide services to people over 65 years of age with functional dependency. The central government partly finances these services alongside co-funding by regional governments and copayment by the person eligible to receive the care. Policies, funding, and standards concerning LTCHs are decided by regional governments. This public system of care co-exists with a private long-term care market. Prior to the pandemic, regional governments did not monitor the availability or adequacy of personal protection equipment (PPE) in LTCHs, LTCH staff were not sufficiently trained to prevent and control infectious diseases, and epidemiological surveillance in LTCHs was very limited [13]. This general lack of preparedness created a fertile ground for the spread of COVID-19 in LTCHs. As a result, COVID-19 deaths in LTCHs accounted for more than 40% of all COVID-19 deaths in Spain during the first year of the pandemic [14].

In the Community of Madrid, LTCHs are under public or private ownership, which can be further classified under nonprofit or for-profit ownership types. Public LTCHs are owned by the regional government or, less frequently, by a municipality. Nonprofit LTCHs are mostly owned by religious orders affiliated with the Catholic Church or, in a few instances, by cooperatives or foundations. For-profit LTCHs are owned by local small and medium enterprises and, increasingly, by international corporations. These corporations have increasingly funded LTCH construction as care for the disabled elderly population is considered a profitable business in rapidly aging countries such as Spain [15].

In addition to ownership, LTCHs can be characterized by their administration type. The administration of publicly owned LTCHs may be either public or operate under a public–private partnership (PPP) agreement. During the last 25 years, the Government of Madrid has actively pursued the PPP model. PPPs have two possible modalities. The first is that a private enterprise manages the staff and provides all services to residents in a publicly owned LTCH as per a three- to five-year contractual arrangement with the Government of Madrid. This completely private administrative modality, consisting of procurement to a private enterprise, has been called "indirect management" by the Government of the Community of Madrid and will henceforth be referred to as "indirect administrative management". The second is that a privately owned LTCH rents a number of places to the government on a contractual basis. Furthermore, there is a completely private sector, where LTCHs are fully owned and managed by private companies or nonprofit organisations and are totally funded by residents through user fees or mixed sources of funding.

A mortality study on a cohort of residents, which started in 1998–1999 in the Community of Madrid, found that public and subsidised LTCHs had higher all-cause mortality than private LTCHs [16]. The mortality rate differences were large and were partially mediated by the larger size of public and subsidised homes. At the end of the 20th century, most private LTCHs were local family businesses and investment in the sector by large corporations was barely nascent.

Our aim was to assess whether LTCH ownership and administration type were associated with mortality during the first two months of the COVID-19 pandemic in the Community of Madrid (Spain).

## 2. Methods

In this cross-sectional study of the registered LTCHs in the Community of Madrid, the outcome variable was the all-cause mortality of residents, estimated by the number of deaths divided by the number of authorised places at each LTCH. We analyse all-cause mortality rather than COVID-19 mortality as the information on the cause of death for deaths that took place in hospital settings is not available.

The all-cause mortality data at LTCHs in the Community of Madrid were obtained from the Portal de Transparencia (Transparency Portal), a government office which provides access to information of public interest upon request. We include the data from 470 out of the 475 licensed LTCHs; five private nursing homes were excluded because their mortality data were unavailable.

The main explanatory variable, LTCH type, is defined in terms of both ownership and administration. There are eight categories of LTCH type, including various combinations of ownership type (for-profit, nonprofit, or owned by the regional or municipal governments) and administration type (public, indirect administrative management, private with places rented by the public sector, or private). Table 1 shows the distribution of 470 LTCHs in the Community of Madrid according to the resulting eight types.

**Table 1.** LTCHs by ownership and administration type in the Community of Madrid, March–April 2020.

| Administration | Ownership | | | |
|---|---|---|---|---|
| | Private (*n* = 405) | | Public (*n* = 65) | |
| | For-Profit (*n* = 329) | Nonprofit (*n* = 76) | Municipal (*n* = 22) | Regional (*n* = 43) |
| Private with rented places | 135 | 19 | | |
| Indirect administrative management | | | 20 | 18 |
| Private | 194 | 57 | | |
| Publicly managed | | | 2 | 25 |

There were 329 for-profit LTCHs, 135 of which rented places to public authorities and 194 of which were under full private administration. There were an additional 76 privately owned non-profit homes, 19 of which rented places to the public sector; the remaining 57 were fully privately financed. Lastly, 65 homes were publicly owned; 22 of them belonged to municipalities and 43 to the Government of the Community of Madrid. Of the municipal LTCHs, 20 were under indirect administrative management (managed by private firms under contracts of limited duration), and 2 were owned and managed by the municipal government. Of the regional LTCHs, 18 were under indirect administrative management, and the remaining 25 were managed by the regional government of the Community of Madrid. Therefore, out of the 470 LTCHs, 65 were publicly owned but only 27 were under public administration. Out of the total 51,938 places, 6258 were in publicly owned and administered LTCHs, 13,308 were in for-profit privately owned and administered LTCHs, 4001 were in non-profit privately owned and administered LTCHs, and the remaining 28,299 were in PPPs (types 1, 2, 3, and 4).

Table 2 shows the facility size by LTCH type in March 2020. Public LTCHs were the largest while privately owned and administered LTCHs, either for-profit or non-profit, were smaller. Municipal LTCHs were the smallest.

**Table 2.** Descriptive statistics of LTCH size by LTCH type in the Community of Madrid.

| | LTCH Type | Facility Size | | | |
|---|---|---|---|---|---|
| | | Mean (SD) | Median | Maximum | Minimum |
| 1. | Profit-based with rented places (*n* = 135) | 161 (71) | 162 | 456 | 40 |
| 2. | Nonprofit with rented places (*n* = 19) | 126 (75) | 100 | 320 | 46 |
| 3. | Municipal, indirect administrative management (*n* = 20) | 71 (40) | 61 | 134 | 13 |
| 4. | Regional, indirect administrative management (*n* = 18) | 148 (58) | 148 | 220 | 52 |
| 5. | Profit-based ownership and administration (*n* = 194) | 69 (64) | 47 | 495 | 10 |
| 6. | Nonprofit ownership and administration (*n* = 57) | 70 (46) | 56 | 202 | 13 |
| 7. | Regional, publicly owned and administered (*n* = 25) | 250 (174) | 201 | 604 | 48 |
| 8. | Municipal, publicly owned and administered (*n* = 2) | 36 (11) | 36 | 44 | 28 |

Figure 1 shows the distribution of LTCHs and places according to the seven LTCH types. The LTCH type is the main exposure variable in this research. Notice that we have excluded the two municipal publicly owned and administered homes (Type 8) because of insufficient sample size for further statistical analysis. The first chart in Figure 1 shows that 41% of the LTCHs in the Community of Madrid were administered by PPPs (types 1, 2, 3, and 4), 53% were privately owned and administered (types 5 and 6), and only 5% were owned and administered by the Government of the Community of Madrid. The second chart shows that 54% of all places were in PPPs (types 1, 2, 3, and 4), 34% were in privately owned and administered homes (types 5 and 6), and only 12% were in publicly owned and administered homes.

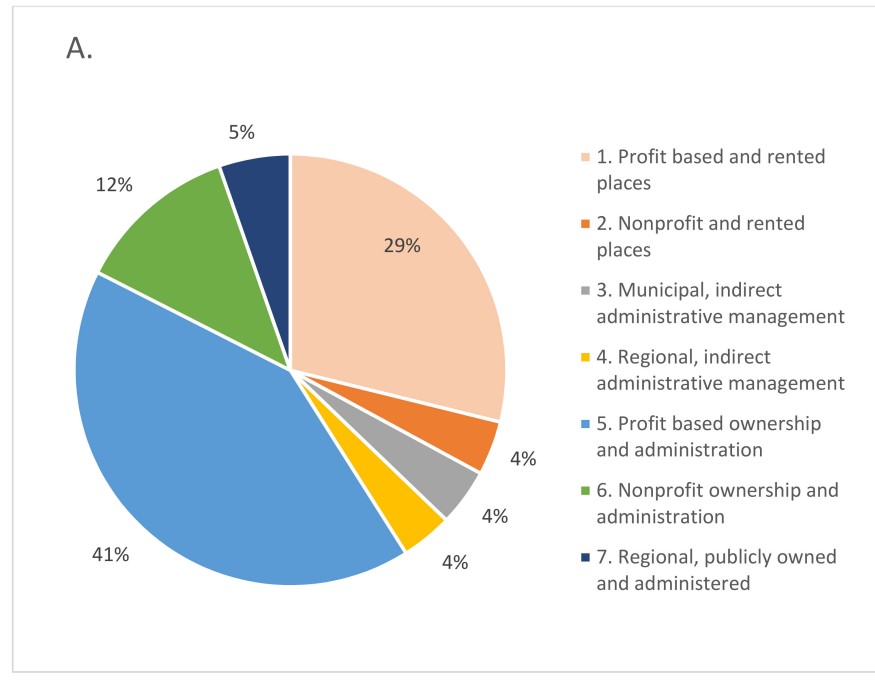

**Figure 1.** *Cont.*

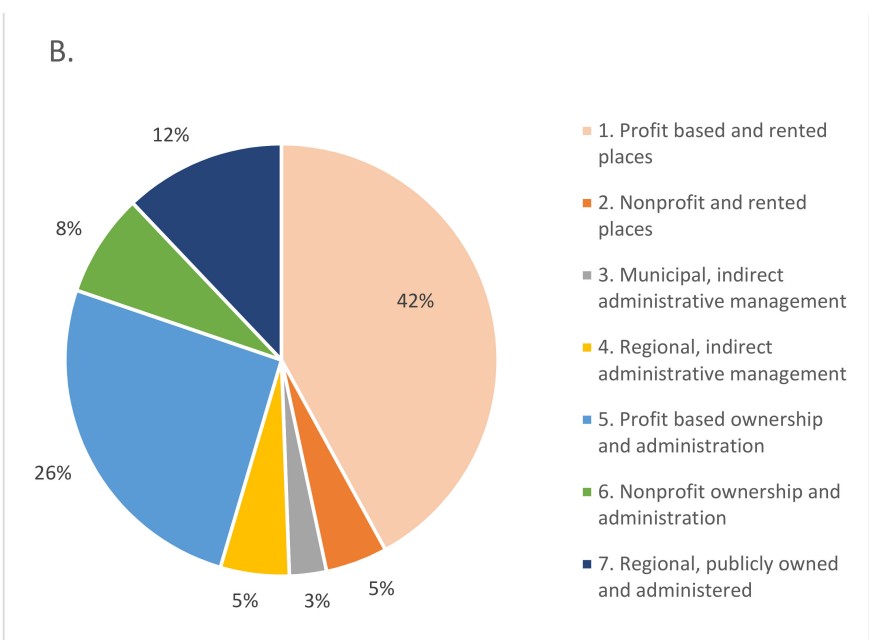

**Figure 1.** LTC homes and places by ownership and management type in the Community of Madrid. (**A**) Percentage of LTCHs by type. (**B**) Percentage of places by LTCH type.

Potential confounders. According to the available literature we consider the following variables to be potential confounders of the association between mortality and LTCH type: size [3,17] and location [3,18].

LTCH size. Prior to the pandemic, the LTCH occupancy rate was close to 100% [19]. Therefore, the number of occupied places approximated the LTCH size. According to current legislation in the Community of Madrid, LTCHs with more than 49 places must have an infirmary unit. For homes with more than 99 places, infirmary beds must equal at least 5% of the total number of places, and there must be a mortuary and an area to provide special services to residents, such as social work and rehabilitation. As the size of the LTCH impacts staffing numbers and the services provided, we have categorised the LTCH sizes as follows: 0–49, 50–99, 100–199, and 200+ places.

LTCH location. The local incidence of COVID-19 is a strong risk factor for mortality in LTCHs [17,20]. During March and April 2020, SARS-CoV-2 infections surged in the city of Madrid and its metropolitan area. Smaller cities and villages started reporting cases later. Although the geographic area incidence of SARS-CoV-2 infection is not available in the Community of Madrid for these two months, we consider population size in the town or city where the LTCH was located as a proxy for cumulative COVID-19 incidence in the area as the first cases were detected in the city of Madrid, then in the metropolitan area, and lastly in the nearby towns. The distribution was as follows: 145 LTCHs in towns with less than 20,000 inhabitants, 51 LTCHs in towns with between 20,000 and 50,000 inhabitants, 62 LTCHs in towns with between 50,000 and 100,000 inhabitants, 55 LTCHs in towns or cities with between 100,000 and 250,000 inhabitants, and 157 LTCHs in the city of Madrid.

Intermediary variable. Staff and visitors who acquired infection in the community could transmit infection to residents and co-workers. The larger the number of employees and residents, the larger the pool of susceptible people was and thus the larger the spread of the SARS-CoV-2 infection. Large LTCHs were either public, belonged to a religious order, or were owned by an enterprise running a chain of homes, often under PPP agreements. Therefore, the spread of the SARS-CoV-2 infection may be an intermediary variable in the association between LTCH type and mortality.

The spread of the SARS-CoV-2 infection in an LTCH was estimated by the number of residents with PCR or antigen tests confirming active infection. During March and April 2020, the number of mean positive cases per LTCH was 22 (SD = 30) and ranged

from 0 to 183. Although 123 LTCHs reported zero positive cases, 17 of them had at least one confirmed or suspected COVID-19 death. This anomaly is likely due to the lack of diagnostic tests at the early stage of the pandemic. For these 17 homes, the number of positive cases was imputed by the number of confirmed or suspected COVID-19 deaths as this was effectively the lower limit for the number of positive cases at those LTCHs.

Nesting of LTCHs in referral hospitals. Hospital care for LTCH residents is provided by the closest public hospital, called the referral hospital. Each referral hospital is responsible for providing healthcare for all the LTCHs in its catchment area (Table 3). In the absence of the cumulative incidence of SARS-CoV-2 infection data in March and April 2020, a hospital's ability to withstand overload reflected the strength of the pandemic in the area.

**Table 3.** Percentage of total deaths occurring after transfer to the hospital during March–April 2020.

| Hospital Name | Number of LTCHs | Percentage of Total Deaths Occurring at the Hospital |
|---|---|---|
| Gómez Ulla | 2 | 13.2 |
| Hospital del Sureste | 12 | 15.5 |
| Infanta Cristina | 13 | 16.0 |
| Villalba | 15 | 16.6 |
| Príncipe de Asturias | 15 | 18.6 |
| Henares | 5 | 19.3 |
| Fundación Alcorcón | 4 | 19.4 |
| Ramón y Cajal | 28 | 21.3 |
| El Escorial | 26 | 21.6 |
| Torrejón | 4 | 22.2 |
| Severo Ochoa | 7 | 23.7 |
| La Princesa | 22 | 24.8 |
| Puerta de Hierro | 36 | 25.5 |
| Tajo | 5 | 25.5 |
| Getafe | 9 | 26.6 |
| Móstoles | 6 | 28.8 |
| Fundación Jiménez Díaz | 36 | 29.5 |
| La Paz | 35 | 29.6 |
| Fuenlabrada | 6 | 30.3 |
| Doce de Octubre | 13 | 30.9 |
| Infanta Sofía | 40 | 31.2 |
| Infanta Elena | 15 | 31.5 |
| Clínico de San Carlos | 15 | 32.6 |
| Infanta Leonor | 9 | 33.5 |
| Gregorio Marañon | 10 | 38.4 |
| Rey Juan Carlos | 37 | 41.4 |

*n* = 425 LTCHs with at least one death.

In addition, an executive order was issued by the Ministry of Health of the Community of Madrid on 18 March 2020 instructing all public hospitals to exclude any patients referred by an LTCH from hospitalisation if they had mobility limitations or severe cognitive decline or disability. This executive order was in effect for three weeks and continued to be applied to a lesser degree by the geriatric departments of the 26 Community of Madrid public

hospitals during April 2020 (Table 3). Admission to private hospitals was rarely sought as a small percentage of LTCH residents had private insurance.

Statistical analysis. Data analysis was conducted in three stages. Firstly, the bivariate association between all-cause mortality (number of deaths/number of places) and LTCH type was tested using a chi-squared test. To assess the potential confounding of LTCH size and location and the spread of infection, the bivariate associations between (a) LTCH size and location and the number of positive cases with mortality and (b) LTCH size and location and the number of positive cases with LTCH type were examined.

Secondly, a multiple linear regression of mortality (number of deaths/facility size) was fitted to consider LTCH type and size. We further adjusted by the number of positive cases as a potential mediating variable to estimate the strength of the association between LTCH type and mortality, independently of the extent of the spread of infection.

Thirdly, a multilevel model was fitted to control the referral hospitals. The intraclass correlation coefficient estimated was 0.047 ($p < 0.05$). The multilevel model included the random hospital intercept and the effects of LTCH type, size, and number of positive cases. Multiple comparisons of group means were tested using the Bonferroni method and took publicly owned and administered homes as the reference category.

## 3. Results

### 3.1. Descriptive Analysis of Mortality

In the Community of Madrid, the number of people living in LTCHs in March 2020, estimated by the number of authorised places, was 52,292. There were 51,938 places in the 470 LTCHs with mortality data. The mean size of an LTCH was 110.5 places (SD = 90.3), and the range was from 10 to 604 places.

During March and April 2020, 9468 deaths among the residents of the 470 LTCHs were reported; 7290 of those deaths occurred at the LTCH and 2178 at the corresponding referral hospital, totalling a mortality of 18.3% in these two months, if full occupancy is assumed. Of the 7290 deaths that occurred in the LTCHs, 1118 were confirmed as COVID-19 deaths, 4676 as deaths with COVID-19 symptoms, and 1496 as deaths due to other causes. The cause of death was not available for those who died after transfer to the referral hospital.

At the LTCH level, the average mortality was 15.8% (median: 15.1; interquartile range: 6.1–23.5%). The maximum number of deaths that occurred in a single LTCH was 113, and these occurred in the largest home with 604 places. The highest percentage of mortality (56%) was observed in a small home with 34 places. Ten percent of the homes experienced more than 50 deaths, whereas 45 LTCHs did not report any deaths in those two months.

Mortality increased with LTCH size: 10% for homes with less than 50 places, 15% for those with between 50 and 99 places, 21% for those with between 100 and 199 places, and 18% for those with 200 places and more (linear test for trends *p*-value < 0.001).

Figures 2–4 show the observed mortality of residents in the LTCHs of the Community of Madrid in March and April 2020. The observed mortality varied by LTCH type. In the LTCHs under PPP agreements, mortality ranged between 18% in the nonprofit LTCHs with rented places and 23% in the LTCHs owned by the Government of the Community of Madrid under indirect administrative management (Figure 2).

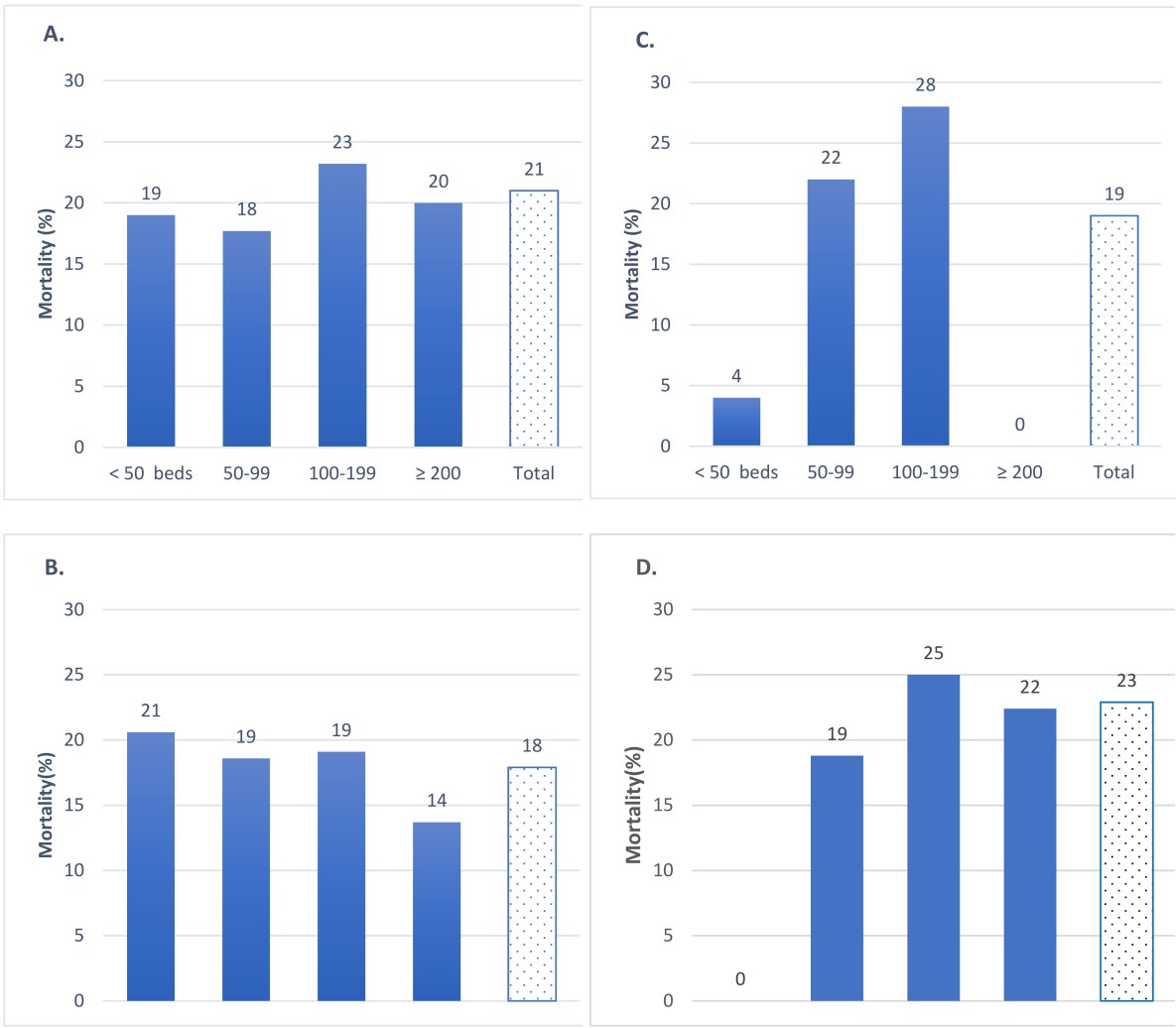

**Figure 2.** All-cause mortality by LTCH size in LTCHs under PPPs, Community of Madrid, March–April 2020. (**A**) For profit with rented places (**B**) Nonprofit with rented places (**C**) Municipal LCTHs with indirect administrative management (**D**) Regional LCTHs with indirect administrative management.

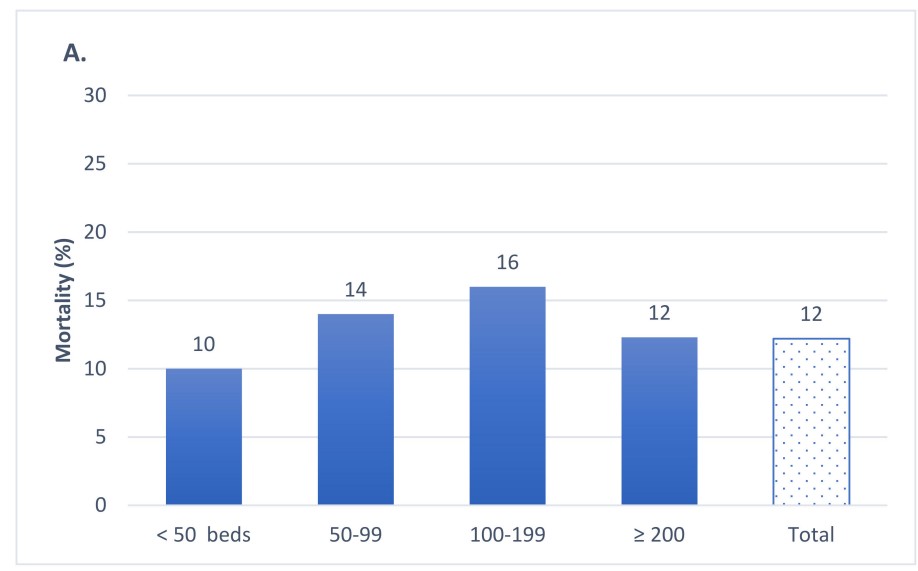

**Figure 3.** *Cont.*

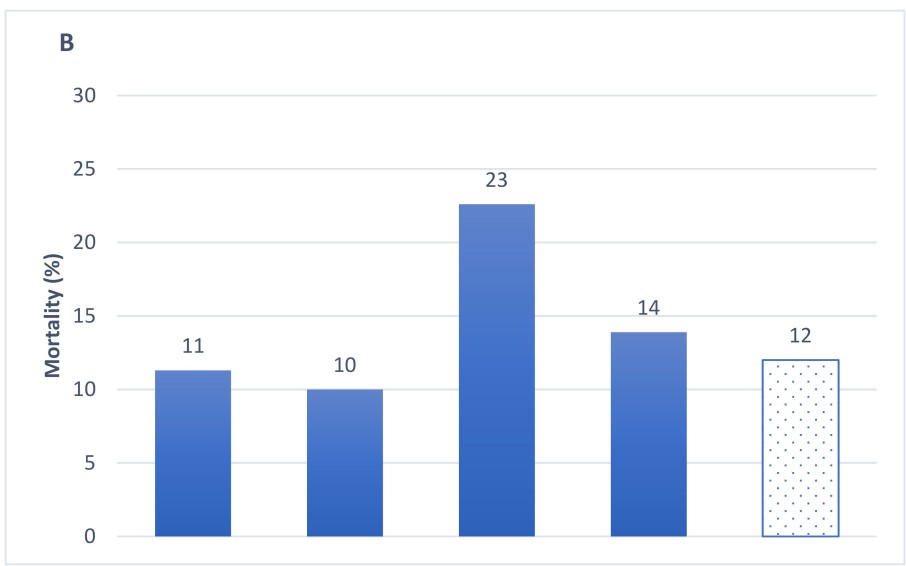

**Figure 3.** All-cause mortality by LTCH size in privately owned and administered LTCHs, Community of Madrid, March–April 2020 (**A**) For-profit (**B**) Nonprofit.

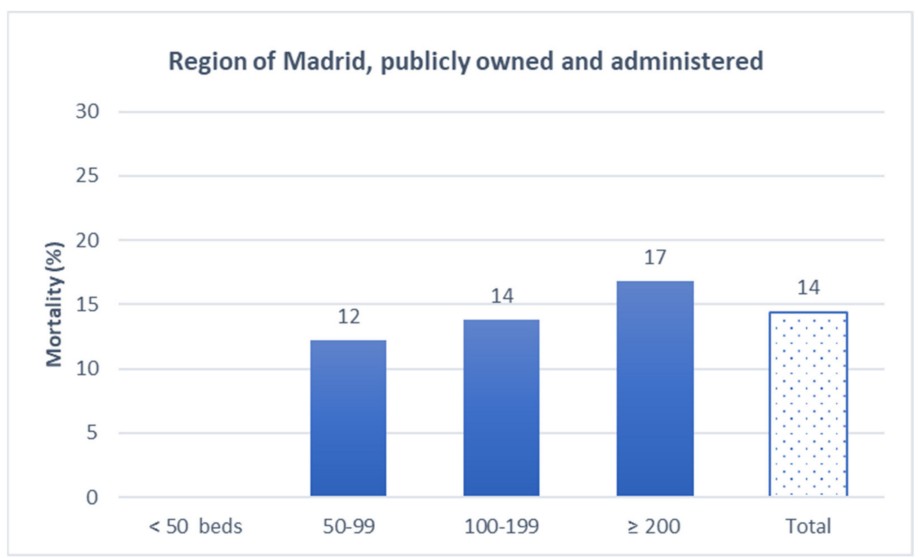

**Figure 4.** All-cause mortality in publicly owned and administered LTCHs, Community of Madrid, March–April 2020.

Both for-profit and nonprofit LCTHs experienced 12% mortality (Figure 3). Mortality in the publicly owned and administered LCTHs was 14% (Figure 4).

### 3.2. Bivariate Analysis to Assess Confounding

LTCH size and location were both significantly associated with LTCH type and mortality, confirming their role as potential confounders. The results of these analyses are available upon request.

### 3.3. Multiple Regression Analysis

We conducted a multiple regression analysis to estimate mortality in the unadjusted and adjusted models, as shown in Table 4. In the unadjusted model, profit-based LTCHs with rented places and indirect administrative management had more than 19% mortality, while the lowest mortality was estimated for types 5, 6, and 7: completely private, either for-profit or nonprofit, and publicly owned and administered.

**Table 4.** Estimated mortality by LTCH type in the Community of Madrid, March–April 2020.

| LTCH Type | Unadjusted | Adjusted by LTCH Size and Location | Adjusted by LTCH Size, Location, and Spread of Infection | Multilevel Model Adjusted by Size, Location, Spread of Infection, and Referral Hospital |
|---|---|---|---|---|
| | Mortality % (95% CI) | Mortality % (95% CI) | Mortality % (95% CI) | Mortality % (95% CI) |
| 1. Profit-based with rented places | 21.5 (19.8–23.2) | 19.5 (17.6–21.4) | 18.8 (17.0–20.6) | 20.6 (18.7–22.5) |
| 2. Nonprofit with rented places | 17.9 (13.4–22.4) | 17.9 (13.5–22.3) | 16.9 (12.6–21.1) | 17.0 (12.7–21.4) |
| 3. Municipal, indirect administrative management | 19.4 (15.0–23.8) | 20.3 (16.0–24.7) | 19.8 (15.6–24.0) | 20.6 (16.4–24.8) |
| 4. Regional, indirect administrative management | 22.9 (18.3–27.5) | 20.3 (15.7–24.9) | 20.3 (15.9–24.7) | 21.9 (17.4–26.4) |
| 5. Profit-based ownership and management | 12.0 (10.8–13.4) | 13.7 (12.0–15.4) | 13.6 (11.9–15.2) | 13.8 (12.0–15.6) |
| 6. Nonprofit ownership and management | 12.3 (9.7–14.9) | 12.6 (9.7–15.5) | 11.7 (8.9–14.5) | 12.7 (10.0–15.4) |
| 7. Publicly owned and administered | 14.5 (10.8–18.4) | 12.9 (9.0–16.8) | 7.4 (3.2–11.6) | 7.4 (3.1–11.7) |

Adjusting by facility size and location provides similar results. Further adjusting by the spread of infection resulted in a drop in the mortality estimated at the publicly owned and administered LTCHs, suggesting that these did better in controlling infections and avoiding mortality compared with the remaining LTCH types.

*3.4. Multilevel Analysis to Consider Referral Hospitals*

The last column of Table 4 shows the estimated marginal means in the multilevel model adjusted by facility size, the number of positive residents, and the referral hospital. Compared with publicly owned and administered LTCHs, LTCH types 1 through 5, private (for-profit or nonprofit) with rented places, indirect administrative management (regional or municipal) and for-profit fully private, had significantly higher mortality.

Figure 5 shows results of a multiple comparison analysis, which took publicly owned and administered LTCHs as a reference point. Four home types under PPP agreements had significantly higher mortality after Bonferroni correction. Significant excess mortality was estimated for types 1, 2, 3, 4, and 5. Mortality in nonprofit private LTCHs was not significantly different from publicly owned and administered homes.

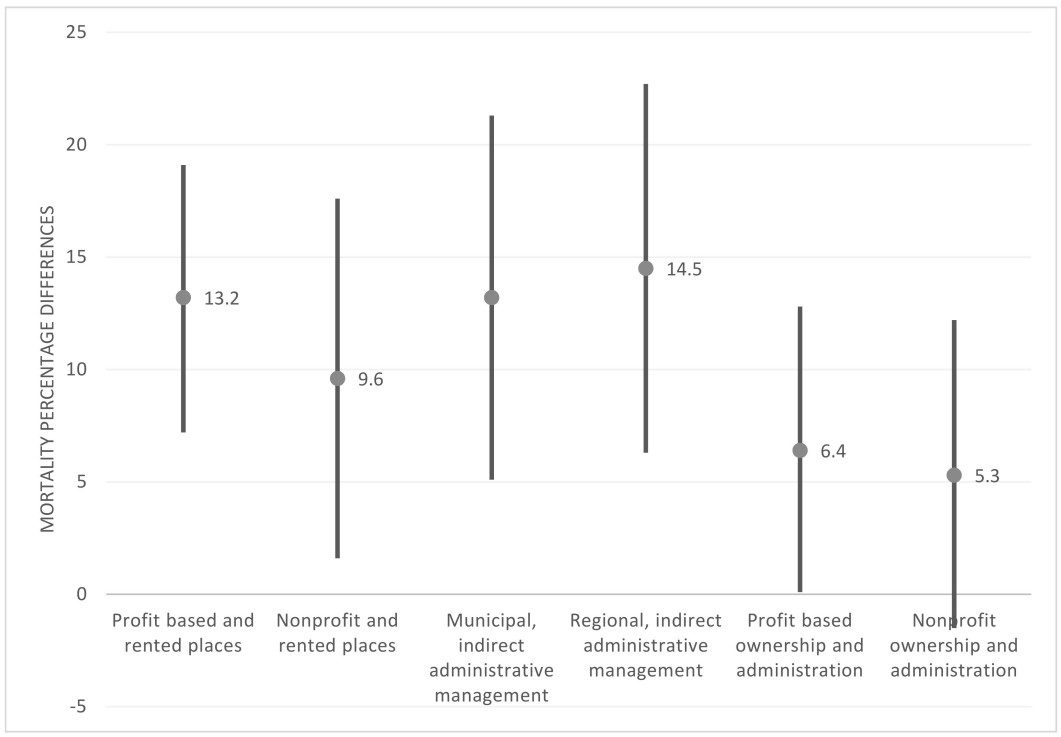

**Figure 5.** Mortality percentage differences between each LTCH type and the publicly owned and administered type, adjusted by size, location, spread of infection, and hospital clustering (95% CI).

Both regional and municipal LTCHs with indirect administrative management had the largest excess mortality (14.5% and 13.2%, respectively). For-profit LTCHs with rented places had 13.2% excess mortality, followed by the nonprofit LTCHs with rented places (9.6%). Lastly, for-profit fully private LTCHs had significant excess mortality when compared with publicly owned and administered LTCHs (6.4%).

## 4. Discussion

Summary of results. This study reveals that during the first two months of the COVID-19 pandemic in the Community of Madrid the LTCHs under public ownership and administration experienced lower mortality than the LTCHs under PPP agreements. The mortality in the for-profit privately owned and administered LTCHs was higher than in the government-administered public LTCHs. The difference was small yet statistically significant.

The public LTCHs under indirect administrative management experienced an almost three-fold mortality rate relative to the government-administered public LTCHs. The nonprofit privately owned and administered LTCHs experienced a mortality similar to that of the publicly owned LTCHs, but the 95% confidence interval was large.

Adjusting by LTCH size and location produced slight changes in the mortality estimates. After including the number of positive cases in the equation, a drop in the mortality of the publicly owned and administered homes was observed, while the mortality estimates for the other types of LTCHs saw very small changes. This indicates that the publicly owned and administered LTCHs performed a better control of infections and avoided mortality once the virus had entered the home, compared with the other categories of LTCHs.

Lastly, adjusting for the nesting of LTCHs in a public referral hospital did not change the estimates of the associations between LTCH type and mortality. This means that even if the degree of exclusion protocol application varied across referral hospitals, it did not confound the association of LTCH type and mortality.

Interpretation given the available literature. These results have broad implications as 41% of LTCHs in the Community of Madrid operated under PPP agreements (categories 1, 2, 3, and 4), caring for 54.5% of all residents. Type 1 alone, for-profit with places rented by the regional government, cared for 42% of all residents. Our results agree with previous research conducted in Ontario, Canada, where long-term care is contracted out through a mixed welfare model that includes non-profit, for-profit, and government service provision. In Ontario, which is among the Canadian provinces with the largest long-term care sector privatised through contracting, "government-run LTCHs outperformed for-profit LTCHs in their management of the COVID-19 pandemic" [5]. More specifically, Pue et al. (2021) reported that COVID-19 mortality was 1.3% in government-administered LTCHs, 3.4% in LTCHs administered by nonprofit organisations, and 5.6% in for-profit LTCHs.

In Spain, as in many high-income countries, social services are being increasingly privatised by contracting out services, often through PPP agreements. Arguments for and against privatisation can be found in Pue et al. (2021), which demonstrated that, at least in Ontario, the government-administered LTCHs had lower COVID-19 infection prevalence and COVID-19 mortality than the for-profit LTCHs. It is essential to consider the context of LTCH organisation and provision in order to accurately interpret our results for the Community of Madrid. In all of Spain and particularly in the Community of Madrid, long-term care services have been chronically underfinanced for many years [21,22]. At the peak of the pandemic in March and April 2020, private LTCH providers received a base amount of 53 euros per day, or around 1600 euros per month, for providing accommodation, nutrition, and daily services to a resident in their care. For comparison, the per diem amount was close to 90 euros in the Basque Autonomous Community [22].

Furthermore, and as previously reported in other countries, for-profit LTCHs had lower staff-to-resident ratios and provided lower wages compared with similar homes in the public sector [4,23]. More broadly, women with low qualifications frequently filled LTCH jobs in the private sector in Spain; staff turnover was high and the training and accreditation of workers was neglected, leading to poor quality of care. LTCHs under

indirect administrative management were usually contracted for only a few years at a time, and government inspections were fragmented and infrequent. In the cases of breaches of contract or other labour or quality-related irregularities, the penalties were small. As an illustration, before the pandemic, thirteen of the eighteen companies contracted under the designation of "indirect management" by the Community of Madrid had been found guilty of not fulfilling the terms of their contracts by the judiciary system and ordered to pay fines. All of them except one had their contracts renewed [24]. For-profit large enterprises, which rent places to the government, are by definition seeking profit to distribute to owners and shareholders, and in pre-pandemic times, they had higher mortality and hospitalisation rates than the non-profit LTCHs [25]. The results suggest that those homes were particularly poor at responding to the pandemic, and as a consequence, they had high mortality.

By contrast, non-profit LTCHs—mostly administered by Catholic orders and cooperatives—tend to reinvest their earnings into their homes and into maintaining acceptable staff levels. In personal interviews with nonprofit LTCH employees at one religious congregation and at one cooperative in Madrid, they explained that during the first weeks of the COVID-19 pandemic, they anticipated the severity of the crisis and mobilised internal resources to buy PPE and diagnostic tests, adapted the homes appropriately, and stratified residents by risk of infection. Among publicly owned LTCHs, those which are publicly administered have historically relied on higher operational budgets, more unionised workers, and more stable and secure job positions due to a higher likelihood of their employees having professional degrees, compared to public LTCHs under indirect administrative management. As a result, non-profit private as well as publicly owned and administered LTCHs may have had more funding available in times of crisis. However, it is not only financing that makes a difference as publicly administered LTCHs offer more stable job conditions, have higher staff ratios, and demand more professional training and experience [5,8,18].

A recent study conducted in the Province of Castellón (Spain) showed that a lower ratio of assistant nurses to residents and cumulative COVID-19 incidence in staff were significantly associated with higher mortality after adjusting for LTCH characteristics [18]. Although quantitative data on staff qualifications and working conditions in the Community of Madrid are unavailable, a recently published, large qualitative study conducted an in-depth analysis of the working conditions in LTCHs in several autonomous communities in Spain, including Madrid [26]. Its content provides evidence to support the above statements. Rico provides an account of long-term residential care in Spain in 2020 [15].

Study limitations. First, this study did not include information on the individual characteristics of LTCH residents. However, we have found no evidence that there is any specific association between residents with certain advanced pathologies or disabilities and LTCHs according to their ownership or administration, except for the fact that privately owned and administered homes (compared with all other types, that is, those under PPP agreements or those which are completely publicly administered) had lower prevalence of comorbidity and functional disability, though all LTCHs had an equal prevalence of dementia [16]. Second, a lack of information on the cause of death of those residents who died in hospital settings prevented us from analysing COVID-19 mortality. However, an estimate of the total COVID-19 deaths is possible as we know that 77% of the total deaths occurred at the LTCHs. As transfers to hospital before dying were mostly determined by the executive order that excluded hospital care due to severe disability, we can assume that the observed 5794 COVID-19 deaths at the LTCHs are 77% of all COVID-19 deaths. Then, the total number of COVID-19 deaths would be 5794/0.77 = 7525 deaths. This would be the lower limit of COVID-19 deaths as it has been reported that all-cause mortality doubled in LTCHs without COVID-19 deaths, probably due to underdiagnosis of the SARS-CoV-2 infection [27]. Third, the number of positive COVID-19 cases as assessed in this study is an underestimation of the spread of the SARS-CoV-2 infection due to the limited availability of diagnostic tests during the first few weeks of the pandemic. Access to testing was limited to symptomatic cases when tests started to be available, and more than half of the infected residents were asymptomatic [28]. Lastly, the data on the staff working conditions and

pandemic preparedness of the LTCHs in the Community of Madrid during those months are not available.

## 5. Conclusions and Recommendations

The results of this study are a testimony to the urgency of LTCH transformation as they demonstrate the fatal consequences that PPP agreements in long-term residential care led to during the first months of the COVID-19 pandemic in the Community of Madrid. The present study suggests that the private administration of public homes increased all-cause mortality during the first two months of the pandemic beyond the effects of facility size, local population figures, the spread of the SARS-CoV-2 infection, and the Community of Madrid executive order excluding disabled elderly residents from hospital care. Our findings agree with previous research conducted in Ontario, Canada and may have broader implications, within and beyond Spain, for the reliance of the public sector on private administration of long-term care.

We join appeals in calling for an official investigation of the events that unfolded in the LTCHs of the Community of Madrid during the COVID-19 pandemic [29]. The Government of Spain should intervene by establishing a public network of LTCHs supported by an information system to allow independent quality control and inspections.

**Author Contributions:** Conceptualisation, M.V.Z. and M.R.; methodology, M.V.Z., F.B. and F.J.G.-L.; statistical analysis, M.V.Z.; writing—original draft preparation; writing—review and editing, M.R., F.B. and F.J.G.-L. All authors have read and agreed to the published version of the manuscript.

**Funding:** This research received no external funding.

**Institutional Review Board Statement:** Ethical review and approval were waived for this study due to epidemiologic surveillance of long-term care home surveillance. No individual data on human beings were used.

**Informed Consent Statement:** This research used data aggregated at the long-term care home level. Individual data were not used.

**Data Availability Statement:** The data are available from the first author upon request.

**Acknowledgments:** We thank two LTCHs managers and many LCTH staff and family members of LTCH residents for their time and effort in interviews which helped us to elaborate the conceptual framework for the analyses. We also thank Claudia Loughran for her work as English editor of the manuscript.

**Conflicts of Interest:** The authors declare no conflict of interest.

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
