# Peer review of "The Impact of Long-Term Care Home Ownership and Administration Type on All-Cause Mortality from March to April 2020 in Madrid, Spain"

_epidemiologia, doi:10.3390/epidemiologia3030025_

Round 1
Reviewer 1 Report
Dear Authors,
the manuscript is interesting and needs minor revisions. Here are some suggestions:
-Please structure the abstract better.
-Please check that the KeyWords are MeshTerms.
-Please in Introduction show in more depth the LTCHs and the various differences between the categories. The comparison seems to be a bit superficial for an international reader.
-The methodology is adequate, the statistical analysis is appropriate.
-Figure 2 is a bit confusing, would it be possible to make it simpler?
-The discussion is adequate and the limitations of the study are well represented.
-In the conclusions if possible show how the government could intervene to improve the situation.
- Please check your English.
Please answer point by point.
Kind regards.
Author Response
Reviewer 1. Thank you very much for your positive assessment. We appreciate your comments which have been useful to improve the manuscript. We have followed your suggestions and carried out the following changes, most of them consequences of the MESH keywords chosen.
-Please structure the abstract better.
Done. The word management has been avoided since it is not specific for administration and organisation.
Please check that the KeyWords are MeshTerms.
We have now changed the terms long-term care for long term care, and “administration and organisation” for management.
-Please in Introduction show in more depth the LTCHs and the various differences between the categories. The comparison seems to be a bit superficial for an international reader.
Three paragraphs in introduction have been changed as follows:
In Spain, public universal coverage of long-term care was established in 2006 when legislation to provide long-term care was passed in order to provide services to people over 65 years of age with functional dependency. The central government partly finances these services alongside co-funding by regional governments and copayment by the person eligible to receive the care. Policies, funding, and standards concerning LTCHs are decided by regional governments. This public system of care co-exists with a private long-term care market. Prior to the pandemic, regional governments did not monitor the availability or adequacy of personal protection equipment (PPE) in LTCHs, LTCH staff were not sufficiently trained to prevent and control infectious diseases, and epidemiological surveillance in LTCHs was very limited (13). This general lack of preparedness created a fertile ground for the spread of COVID-19 in LTCHs. As a result, COVID-19 deaths in LTCHs accounted for more than 40% of all COVID-19 deaths in Spain during the first year of the pandemic (14).
In the Community of Madrid, LTCHs are under public or private ownership, which can be further classified under nonprofit or for-profit ownership types. Public LTCHs are owned by the regional government or, less frequently, by a municipality. Nonprofit LTCHs are mostly owned by religious orders affiliated with the Catholic Church or, in a few instances, by cooperatives or foundations. For-profit LTCHs are owned by local small and medium enterprises and, increasingly, by international corporations. These corporations have increasingly funded LTCH construction as care for the disabled elderly population is considered a profitable business in rapidly aging countries such as Spain (15).
In addition to ownership, LTCHs can be characterized by their administration type. The administration of publicly owned LTCHs may be either public or operate under a public-private partnership (PPP) agreement. During the last 25 years the Government of Madrid has actively pursued the PPP model. PPPs have two possible modalities. The first is that a private enterprise manages the staff and provides all services to residents in a publicly owned LTCH as per a three to five-year contractual arrangement with the Government of Madrid. This completely private administrative modality, consisting of procurement to a private enterprise, has been called “indirect management” by the Government of the Community of Madrid and will henceforth be referred to as “indirect administrative management”. The second is that a privately owned LTCH rents a number of places to the government on a contractual basis. Furthermore, there is a completely private sector, where LTCHs are fully owned and managed by private companies or nonprofit organisations and are totally funded by residents through user fees or mixed sources of funding.
-The methodology is adequate, the statistical analysis is appropriate.
Thank you for your appreciation.
-Figure 2 is a bit confusing, would it be possible to make it simpler?
We have replaced figure 2 by figures 2, 3 and 4 and modified the corresponding text.
-The discussion is adequate and the limitations of the study are well represented.
-In the conclusions if possible show how the government could intervene to improve the situation.
We have added the following sentence to the conclusion
The Government of Spain should intervene by establishing a public network of LTCHs supported by an information system to allow independent quality control and inspections.
The new text has been revised by an English editor.
Reviewer 2 Report
1.- "Access to testing was limited to symptomatic cases"
Sanford says. In pandemic in poor countries, if you don´t have tests, but the patients present clinical picture of COVID 19, is COVID 19.
2.- "More of than half of infected residents were asymptomatic"
When they detected one case of COVID 19, in long-healthcare, they din´t do the tests to the other residents (inclusive the asymptomatic). Don´t they?
3.- Do you know the % of infected socio sanitary workers, that could bring the infection to the residents of the long-healthcare ?
4.- When a long-healthcare is opens. Is it accredited by the ministry of health? and is there a monitoring?
5.- Is there informal long-healthcare?
6.- If the answer is affirmative. How many?
7.- In conclusions, goes only conclusions. So the last paragraph is incorrect.
8.- In the discussion the results are compared with the bibliography of the world.
Author Response
Reviewer 2. Thank you for your comments and questions. We will try to answer below.
- Sanford says. In pandemic in poor countries, if you don´t have tests, but the patients present clinical picture of COVID 19, is COVID 19.
Spain is classified as a high-income country by the World Bank but there was a lack of pandemic preparedness, and many older adults died without being tested. Clinical suspicion of COVID-19 was taken as COVID-19 death but infected patients were often undiagnosed. During the first two months of the pandemic only older adults with symptoms were isolated and no effort was made to isolate their contacts.
- "More of than half of infected residents were asymptomatic"
When they detected one case of COVID 19, in long-healthcare, they din´t do the tests to the other residents (inclusive the asymptomatic). Don´t they?
Only symptomatic patients were tested.
- Do you know the % of infected socio sanitary workers, that could bring the infection to the residents of the long-healthcare?
No. There were no official figures but we have data on number of positive workers at each LTCH with the same restrictions as for older adults. Symptomatic cases were tested if the occupational health medical services had availability of tests. No efforts were taken to isolate contacts. In addition, often staff worked at several LTCHs but this was not quantified.
- When a long-healthcare is opens. Is it accredited by the ministry of health? and is there a monitoring?
Yes. Centers are accredited by the regional government of Madrid. There are inspections but they are seldom carried out since there is an insufficient number of inspectors and the evaluation protocol is fragmented into a few domains.
- Is there informal long-healthcare?
Yes. Spain is a familialistic society and families take care of disabled older adults as long as they can but many families cannot provide the number of daily hours of caregiving at moderate or severe disability stages. In Spain, only 3.3% of adults over 65 are living in long term care homes but waiting lists are long given the increase in the number of people that have very long lives. Life expectancy was 86 for women and 81 for men before the COVID_19 pandemic.
6.- If the answer is affirmative. How many?
About 85% of care for disabled elderly comes from informal sources, mostly the women of the family.
7- In conclusions, goes only conclusions. So the last paragraph is incorrect.
We intend to write conclusions and recommendations. We have clarified that subtitle.
8.- In the discussion the results are compared with the bibliography of the world.
We have compared the long-term care systems of Spain and Canada since this country has a similar federal structure and a similar long term care system that has been to that of Spain and it has been recently privatized by increasing use of public-private sector partnership agreements. Even comparisons with other high-income countries is difficult since organization and financing are very diverse. See for example an attempt to compare the features of the US and the French long-term care system.(DOTY et al., 2015) . DOTY, P., NADASH, P., & RACCO, N. (2015). Long-Term Care Financing: Lessons from France. Milbank Quarterly, 93(2), 359–391. https://doi.org/10.1111/1468-0009.12125.
To compare the impact of public-private sector partnerships on mortality across countries was beyond the scope of our paper.
Reviewer 3 Report
Congratulations.
The manuscript is of interest to readers.
I suggest a lengthier description of what kind of study would the authors suggest in the future.
Author Response
Thank you for your comments.
I suggest a lengthier description of what kind of study would the authors suggest in the future.
As requested by you and reviewer no. 1 we have included one sentence in the last paragraph of recommendations.
The Government of Spain should intervene by establishing a public network of LTCHs supported by an information system to allow independent quality control and inspections.
We could write much longer but we think it would go beyond our current research results. We consider urgent to set up an information system that will allow to conduct studies on long term care exposures and outcomes. Today the paucity of information on LTC in Spain is overwhelming. At the moment of writing this paper, it is not possible to conduct a mortality study on residents of LCTHs in Spain, much less to conduct a follow up study with relevant outcomes of care. There is practically no quantitative knowledge on working conditions or competence of hired staff or indicators of quality of care such as ulcers, pharmaceutical drugs se, chemical or physical restrictions use, to name a few challenges.
.